# The Development of Catalyst Materials for the Advanced Lithium–Sulfur Battery

**Hong-Jie Zhou [1], Chun-Lei Song [1], Li-Ping Si [2], Xu-Jia Hong [1],\* and Yue-Peng Cai [1],\***

[1] Guangzhou Key Laboratory of Materials for Energy Conversion and Storage, School of Chemistry, South China Normal University, Guangzhou 510006, China; zhjscnu@163.com (H.-J.Z.); sclscnu@163.com (C.-L.S.)

[2] School of Materials Science and Energy Engineering, Foshan University, Foshan 528000, China; lipingsi@fosu.edu.cn

\* Correspondence: hxuscnu@m.scnu.edu.cn (X.-J.H.); caiyp@scnu.edu.cn (Y.-P.C.); Tel.: +86-020-39310383 (X.-J.H. & Y.-P.C.)

**Abstract:** The lithium–sulfur battery is considered as one of the most promising next-generation energy storage systems owing to its high theoretical capacity and energy density. However, the shuttle effect in lithium–sulfur battery leads to the problems of low sulfur utilization, poor cyclability, and rate capability, which has attracted the attention of a large number of researchers in the recent years. Among them, the catalysts with efficient catalytic function for lithium polysulfides (LPSs) can effectively inhibit the shuttle effect. This review outlines the progress of catalyst materials for lithium–sulfur battery in recent years. Based on the structure and properties of the reported catalysts, the development of the reported catalyst materials for LPSs was divided into three generations. We can find that the design of highly efficient catalytic materials needs to consider not only strong chemical adsorption on polysulfides, but also good conductivity, catalysis, and mass transfer. Finally, the perspectives and outlook of reasonable design of catalyst materials for high performance lithium–sulfur battery are put forward. Catalytic materials with high conductivity and both lipophilic and thiophile sites will become the next-generation catalytic materials, such as heterosingle atom catalysis and heterometal carbide. The development of these catalytic materials will help catalyze LPSs more efficiently and improve the reaction kinetics, thus providing guarantee for lithium sulfur batteries with high load or rapid charge and discharge, which will promote the practical application of lithium–sulfur battery.

**Keywords:** lithium–sulfur battery; catalyst; lithium polysulfides

---

## 1. Introduction

With the advantages of ultra-high theoretical capacity of 1675 mAh $g^{-1}$ and energy density of 2600 Wh $kg^{-1}$, as well as environmental friendliness and low cost, lithium–sulfur battery is regarded as a kind of next-generation energy storage systems [1,2]. However, the practical application of lithium–sulfur battery remains challenging because of many technical bottlenecks. First, the poor conductivity of sulfur leads to a great demand of conductive additives in the positive material, which sharply reduces the actual capacity of lithium–sulfur battery. Second, the soluble lithium polysulfides (LPSs) ($Li_2S_n$, $4 \leq n \leq 8$) can be dissolved in the electrolyte, resulting in rapid capacity degradation, low coulomb efficiency, and short cycle life. Finally, due to the uneven deposition of lithium, lithium dendrites are formed, which will penetrate the separators and lead to the short circuit of battery (Figure 1a,b). At present, the effective way to inhibit the shuttle effect is to improve the ability of LPSs capture and increase the conversion speed between LPSs and insoluble short-chain sulfides ($Li_2S_{(2)}$). Therefore, how to design and synthesize catalytic materials with high adsorption and

catalytic function for LPSs has attracted a lot of research. So far, a variety of catalytic materials have been reported to inhibit shuttle effect. Based on this, this review summarizes the research progress of catalysts to improve the adsorption and conversion rate of LPSs, and divides the development of catalytic materials reported so far into three generations based on the catalytic materials and the catalytic effect of LPSs. The development of catalysts has experienced the evolution from the first generation with strong interaction and certain catalysis on LPSs, but low conductivity, to the second generation that combines conductivity and adsorption, and then to the third generation with stronger adsorption, higher conductivity, and catalytic efficiency (Figure 1c–e). Finally, based on the above summary, prospects for rational design of high-efficiency catalyst materials are presented (Figure 1f).

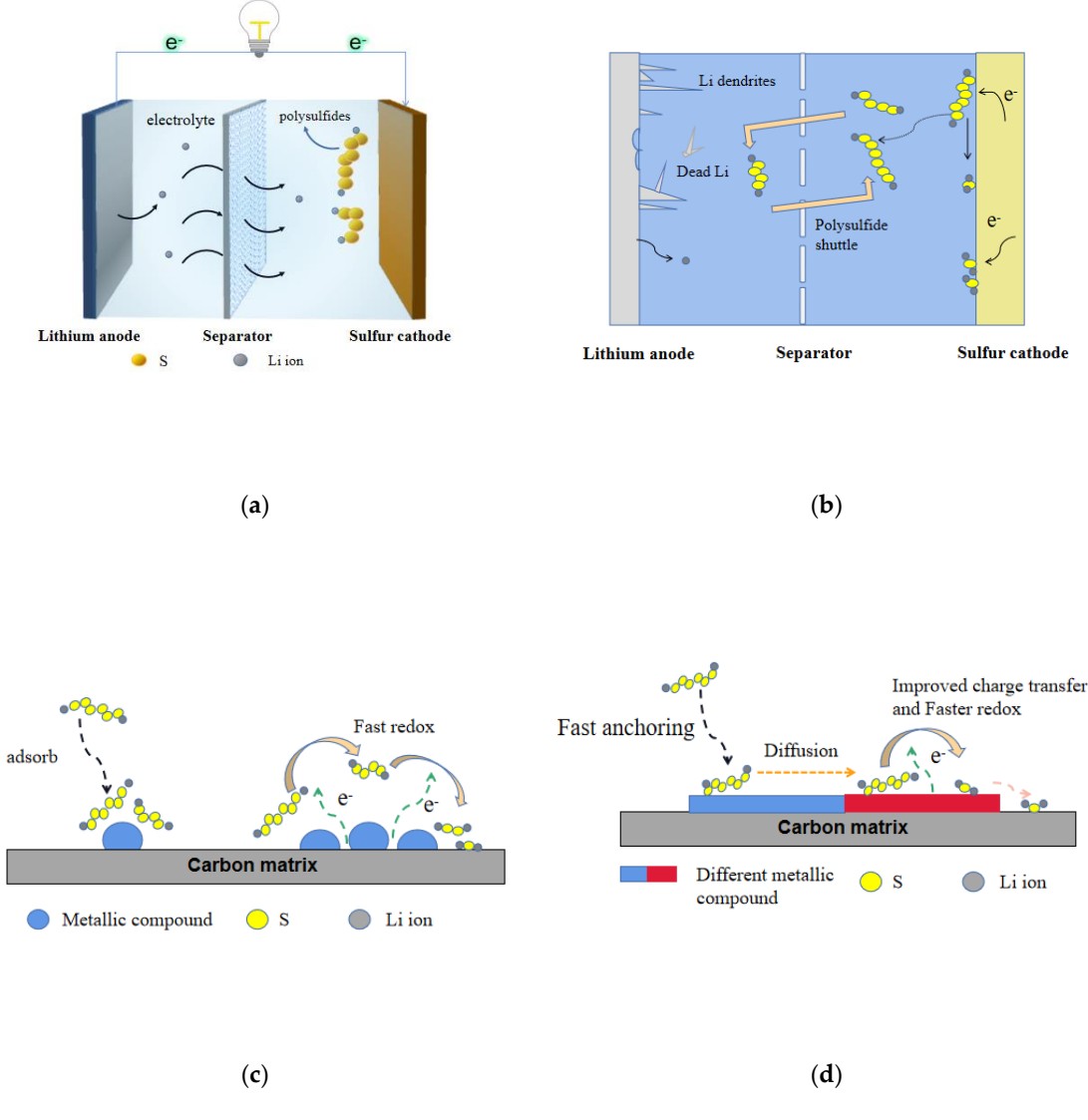

**Figure 1.** *Cont.*

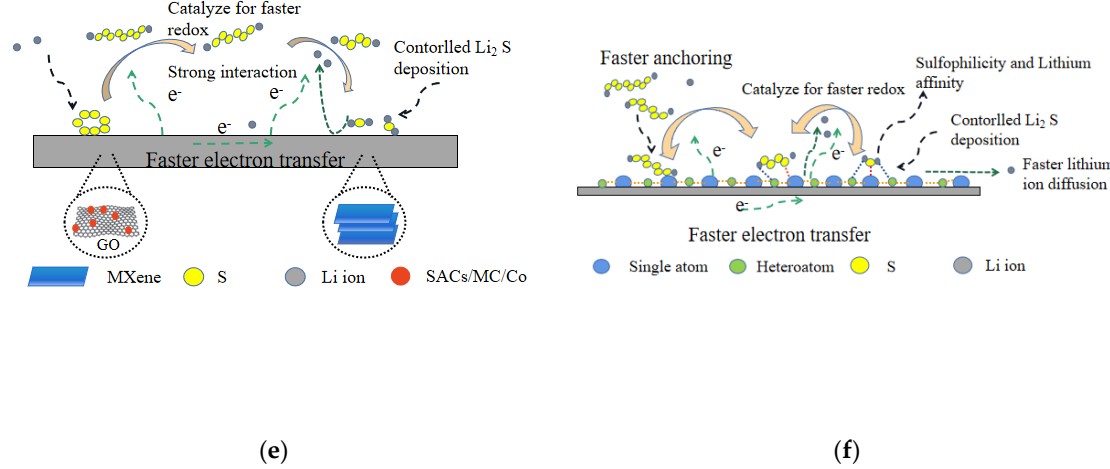

(**e**)                                       (**f**)

**Figure 1.** (**a**) The structure of lithium–sulfur battery. (**b**) Schematic diagram of lithium–sulfur battery challenges. (**c**) The first-generation catalyst materials. (**d**) The second-generation catalyst materials. (**e**) The third-generation catalyst materials. (**f**) The next-generation catalyst materials.

## 2. Before the Catalyst

Before the use of catalyst materials to inhibit the shuttle effect, the reports usually adjust the pore size of the material or choose the material with strong adsorption on LPSs as the sulfur host or separator coating to inhibit the shuttle effect and improve the capacity and cycling performance of lithium sulfide batteries [3–8].

### 2.1. Physical Adsorption

Researchers have made many attempts to inhibit the shuttle effect. For example, some kinds of unique structures were designed, such as mesoporous carbon, active carbon nanotubes, and so on, to restrict S8 and LPSs in the positive electrode [9], or modified the separator by coating with carbon materials, such as graphene oxide (GO) and microporous carbon [10], to prevent the shuttle of LPSs and reuse the adsorbed active materials. However, due to the non-polar surface of carbon materials, the weak interaction between carbon and sulfur or LPSs leads to the fact that these materials cannot effectively adsorb the polysulfides. On this basis, researchers tried various kinds of methods to modify the surface of carbon materials in order to improve its polarity, such as using nitrogen doped micropores carbon, graphene, etc. [11–13]. Yi Jiang and co-workers proposed a self-templating and organic solvent-free method to synthesize monoclinic ZIF-8 nanosheet at room temperature, from which 2D-HPC nanosheet was derived to serve as the host of sulfur cathode (Figure 2a) [14]. The results show that the doping of nitrogen in carbon not only contributes to the rapid charge transfer, but also helps to form a stronger interaction between carbon and LPSs, thus enhancing the immobilization ability on sulfur and polysulfide, making the cell cycle stably. In this method, the cell cycles 300 times at 0.5 C with a capacity decay rate of 0.12% (Figure 2b). Additionally, some reports suggest that the use of small $S_{2-4}$ molecules as the active material can help avoid the shuttle effect caused by the generation of polysulfides during charging and discharging. Thus, we prepared flower-like microporous nitrogen-doped carbon nanosheets with large pore volume and small pore diameter of <0.6 nm using N-rich Metal-Organic Frameworks (MOFs) as a precursor, which is an ideal carrier for small molecules $S_{2-4}$ [15]. The results showed that the capacity after 1000 long-term cycles was 727 mAh g$^{-1}$, the Coulomb efficiency at 2 C was 99.5%, and the capacity loss is only ~0.02% per cycle (Figure 2c). Compared with porous carbon, the pores of MOFs can be modified by chemical active centers, such as Lewis acid center and functional organic groups, so MOFs have a greater ability to block sulfur and polysulfide [16–21]. We combined the Lewis base of the nitrogen atom in the ligand $H_6TDPAT$ with the Lewis acid center of Cu(II) open metal site to form a dual-function cage metal

organic framework Cu-TDPAT as the sulfur host for cathode [22]. DFT calculations and XPS data show that the dual-functional binding site (Lewis acid-base site) in the MOFs, Cu-TDPAT, provides a comprehensive and stable interaction for overcoming the dissolution and diffusion of polysulfides in the electrolyte (Figure 2d,e). At 1 C, the reversible capacity is ~745 mAh g$^{-1}$ after 500 cycles.

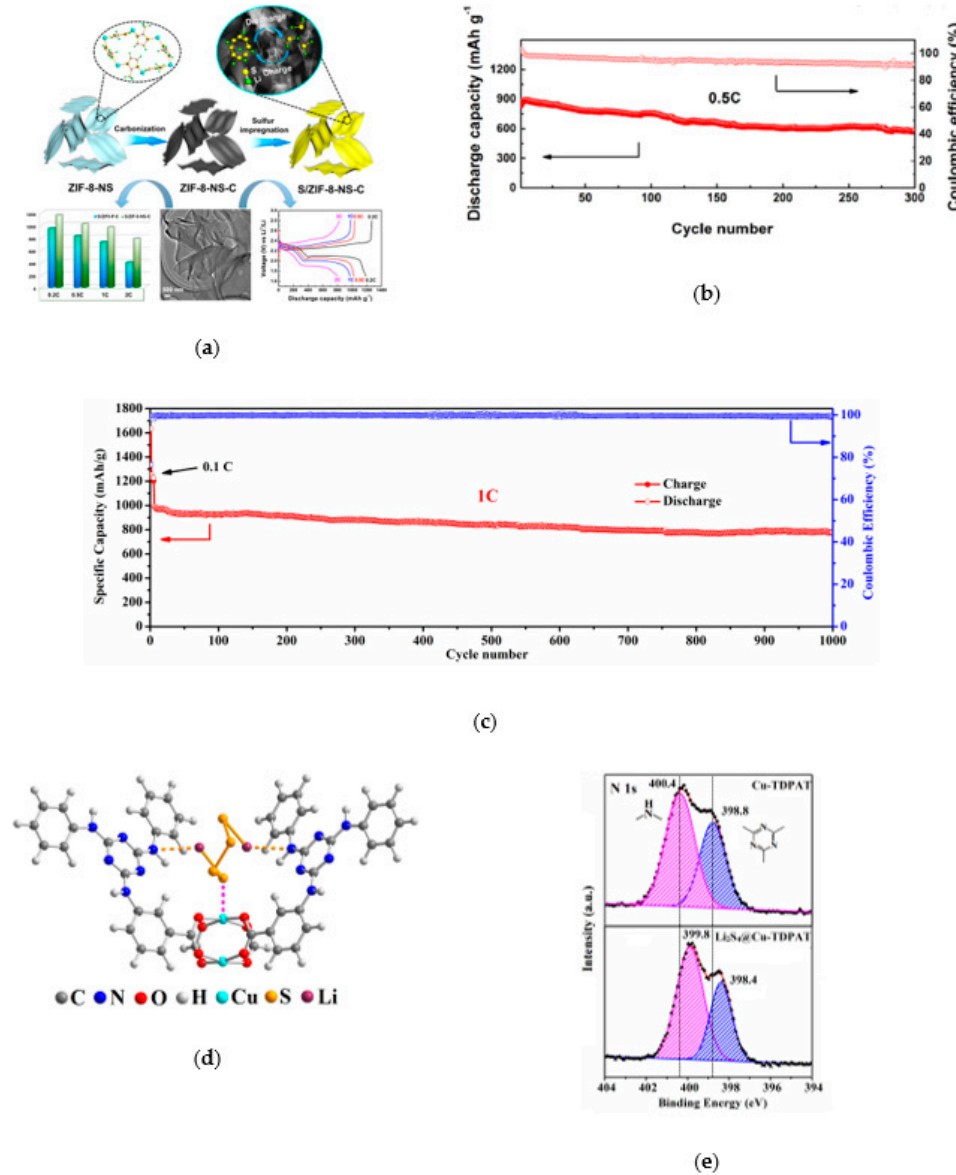

**Figure 2.** (**a**) Schematic illustration of the S/ZIF-8-NS-C composites preparation [14]. (**b**) Long-term cycling performance of cell S/ZIF-8-NS-C at 0.5 C [14]. (**c**) Cycling stability of S/FMNCN-900 cathode at 2 C for 1000 cycles [15]. (**d**) The optimum atomic model configurations showing the interactions between Cu-TDPAT and Li$_2$S$_4$ [22]. (**e**) XPS N1s spectra of Cu-TDPAT and Li$_2$S$_4$@Cu-TDPAT [22].

The adsorption of nitrogen-doped porous carbon or MOFs material to LPSs is weak physical adsorption, which cannot effectively inhibit its shuttle effect. At the same time, the conductivity of the material is also an important factor; the use of poor conductivity of the material will increase the resistance and reduce the rate performance. Therefore, in order to improve the performance of the batteries, it is necessary to select materials with good conductivity and excellent ability to capture LPSs.

### 2.2. Chemical Adsorption

Metal compounds, such as metal oxides [23,24], nitrides [25], phosphides [26], and sulfides [27–30], have strong chemical interaction on LPSs, which have been widely used in lithium sulfur batteries. Yiren Zhong and co-workers found that CoP nanoparticles can strongly adsorb polysulfides (Figure 3a) because of their natural oxidation to form Co-O-Plike species, in which the active Co site on the surface could interact with polysulfides through strong Co–S bond [31]. As the oxide layer on the surface could strongly interact with LPSs, and the inner core is suitable for conducting electrons, the CoP nanoparticles can effectively improve the performance of lithium sulfur battery. With the high sulfur load of 7 mg cm$^{-2}$, the area capacity is 5.6 mAh cm$^{-2}$ after 200 cycles (Figure 3b). Fang Liu and co-workers found that there is a significant correlation between bond energy and cycle stability: the stronger bond energy between oxides and polysulfides, the higher capacity retention rate (Figure 3c) [32]. Shanshan Yao and co-workers prepared 2D-Sb$_2$S$_3$ nanosheet by electrochemical intercalation and stripping [33], and proved that it can effectively capture and recover polysulfide as a coupling material. In addition, as shown in Figure 3d–f, the first principle calculation confirmed that the material has appropriate bond strength of 1.33–2.14 eV, which can capture polysulfide while ensuring the stability of the internal Li-S bond, and the energy barrier of effective diffusion of Li on the surface of SSNs is low.

The adsorption of metal compounds on polysulfides is stronger than that of carbon materials. However, due to the low conductivity of most polar materials, the captured polysulfides stay on the surface of polar materials, which leads to the failure of the captured LPSs to be fully utilized and decrease of the rate performance. Besides, the simple physical or chemical adsorption cannot fundamentally solve the problems of rapid capacity degradation and low coulomb efficiency caused by shuttle effect. Thus, the researchers focused on the catalytic materials with high conductivity, high adsorption, and high catalytic kinetics for LPSs, and applied them to the sulfur host or separator coating, which effectively inhibited the shuttle effect. Herein, we divide the development of catalysts for LPSs into three generations based on the aspects of the adsorption, catalytic of polysulfides, and the conductivity of materials. The characteristics and shortcomings of each generation of catalytic materials are discussed, which provides guidance for the design of the next generation of catalytic materials with better performance.

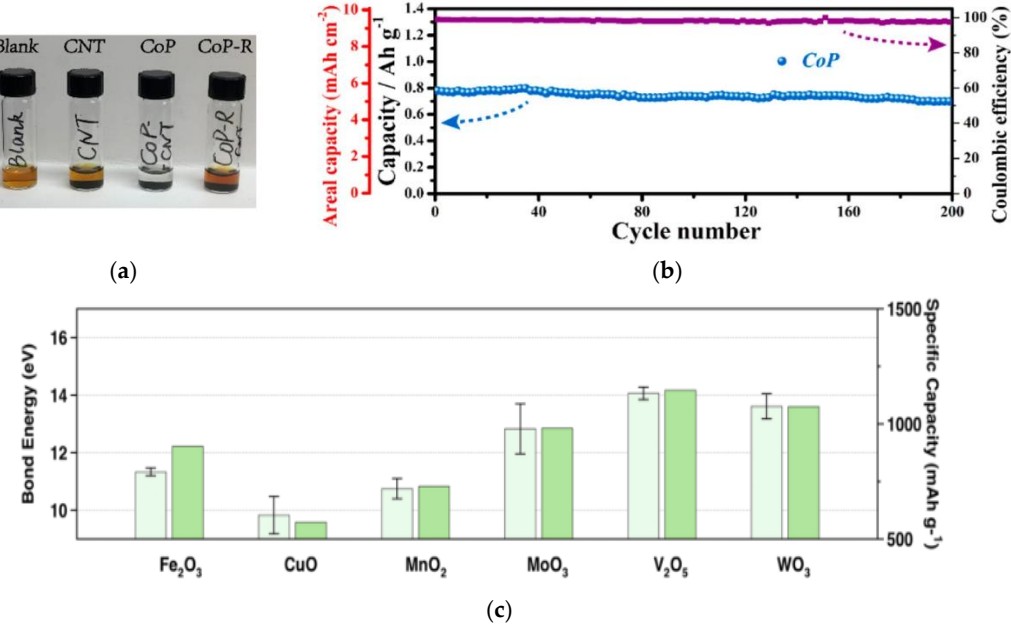

**Figure 3.** *Cont.*

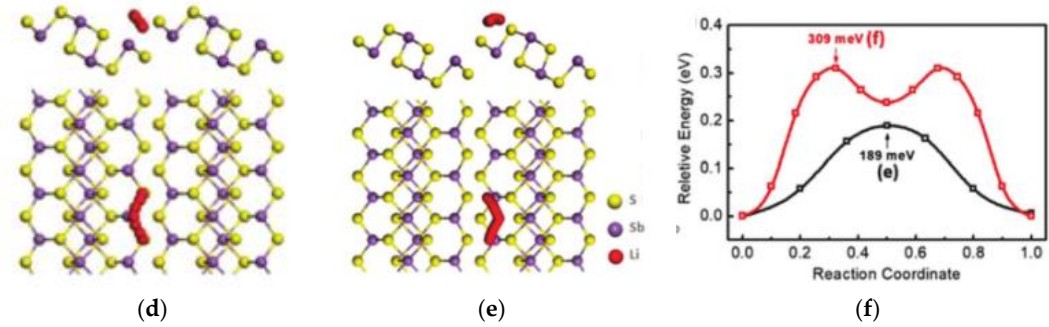

<p style="text-align:center">(<b>d</b>)　　　　　　　　　　　　　　　　(<b>e</b>)　　　　　　　　　　　　　　　　(<b>f</b>)</p>

**Figure 3.** (**a**) Results of LPSs (5 mM $Li_2S_6$-DOL/DME solution) adsorption experiments with Carbon nanotubes (CNTs) and CoP and CoP-R nanoparticles supported on CNTs [31]. (**b**) Cycling performance of a high-capacity S cathode (S mass loading: 7 mg cm$^{-2}$; rate: 0.2 C) modified with CoP-CNT [31]. (**c**) A comparison of the bond energies between the metal oxides and polysulfides (light green) with the specific capacity of the corresponding Li−S cells after 100 cycles at 1 C (green). These cells were made using RSL containing these metal oxides, respectively. The bond energies were calculated with Flore's equation based on dissociation energy, electronegativity, and chemical hardness of metal oxides and polysulfides [32]. (**d,e**) Schematics of top and side views representing two Li diffusion pathways on $Sb_2S_3$ nanosheets and (**f**) the corresponding energy profiles for different diffusion pathways [33].

## 3. The First Generation Catalyst Materials

The slow reduction from long-chain polysulfides to short-chain $Li_2S_{(2)}$ seriously hinders the full utilization of active sulfur. Besides, $Li_2S$, as the final product, has high interface resistance, which requires large activation energy for oxidation. With the in-depth research of lithium sulfur battery, people have found that metal compounds (metal oxides, sulfides, nitrides, phosphates, etc.) not only have a chemical adsorption effect on polysulfides, but also a catalytic effect, which can rapidly adsorb polysulfides and promote their efficient conversion (Figure 1c). This section provides a detailed overview.

### 3.1. Oxides

Metal oxides are the first to be found to have strong chemical adsorption with LPSs, as well as strong catalytic activity for the conversion of polysulfides. Yatao Liu and co-workers used carbon-free porous hollow $NiCo_2O_4$ nanofibers as a host for sulfur to prepared the cathode [34]. As shown in Figure 4a,b, it can be found that $NiCo_2O_4$ nanofibers have strong chemisorption and catalytic ability for polysulfide conversion. The S/$NiCo_2O_4$ with sulfur loading of 1.66 g cm$^{-3}$ provides the capacity of 1867 mAh cm$^{-3}$, almost twice that of the traditional S/carbon composite. As shown in Figure 4c, active nickel and cobalt atoms can provide reaction sites to adsorb polysulfides and catalyze the transformation of soluble LPSs.

Ke Lu and co-workers found that the coupling of polar $Fe_3O_4$ with highly conductive N-doped carbon provides stronger adsorption ability for polysulfides, which is much better than that of N-doped carbon and activated carbon alone [35]. The synergistic effect of $Fe_3O_4$ and N-doped carbon makes the conversion between $Li_2S_4$ and $Li_2S$ more effective, which contributes to high capacity and long-term cycle stability in lithium sulfur battery (Figure 4d). Based on this, Meng Ding and co-workers prepared a kind of three-dimensional $Fe_3O_4$/NC/G ternary composite host for the high rate and capacity lithium sulfur batteries with liquid polyelectrolyte as the positive active material. (Figure 4e) [36]. $Fe_3O_4$ nanocrystals and a three-dimensional interconnection conductive structure are helpful to promote the reaction kinetics, accelerate electron/ion transport, alleviate the nucleation of polysulfide, and improve the rate performance. DFT calculation results clearly show that the bond energy between LPSs and carbon surface (0.10–0.52 eV) is much lower than that between LPSs and $Fe_3O_4$ (0.44–3.35 eV). Besides, the chemical adsorption between polysulfides and $Fe_3O_4$ is mainly due to the stronger polar interaction. The structure of $Fe_3O_4$ and non-polar graphene in $Fe_3O_4$/NC/G is conducive to the combination of

polysulfides and the diffusion of polysulfides on the electrode. Yingze Song designed an innovative in situ synthesis method through plasma-enhanced chemical vapor deposition (PECVD), with defect graphene nanosheet directly grown on the surface of $VO_2$ nanosheet network to obtain $G-V_2O_3$ hybrid (Figure 4f) [37]. In this method, the interface produced by PECVD technology formed a clean and efficient interface between graphene and $V_2O_3$. The Electrochemical Impedance Spectroscopy (EIS) curve showed that the composite has good conductivity and catalytic activity (Figure 4g).

Some metal oxides contribute to promoting the conversion of LPSs, but most of them have poor conductivity, which increases the resistance and reduces the rate capability of the battery. In order to enhance the rate performance and capacity of the battery, it is necessary to improve the conductivity of metal oxides, such as combining oxides with conductive materials, or chose the other metal compounds with better conductivity.

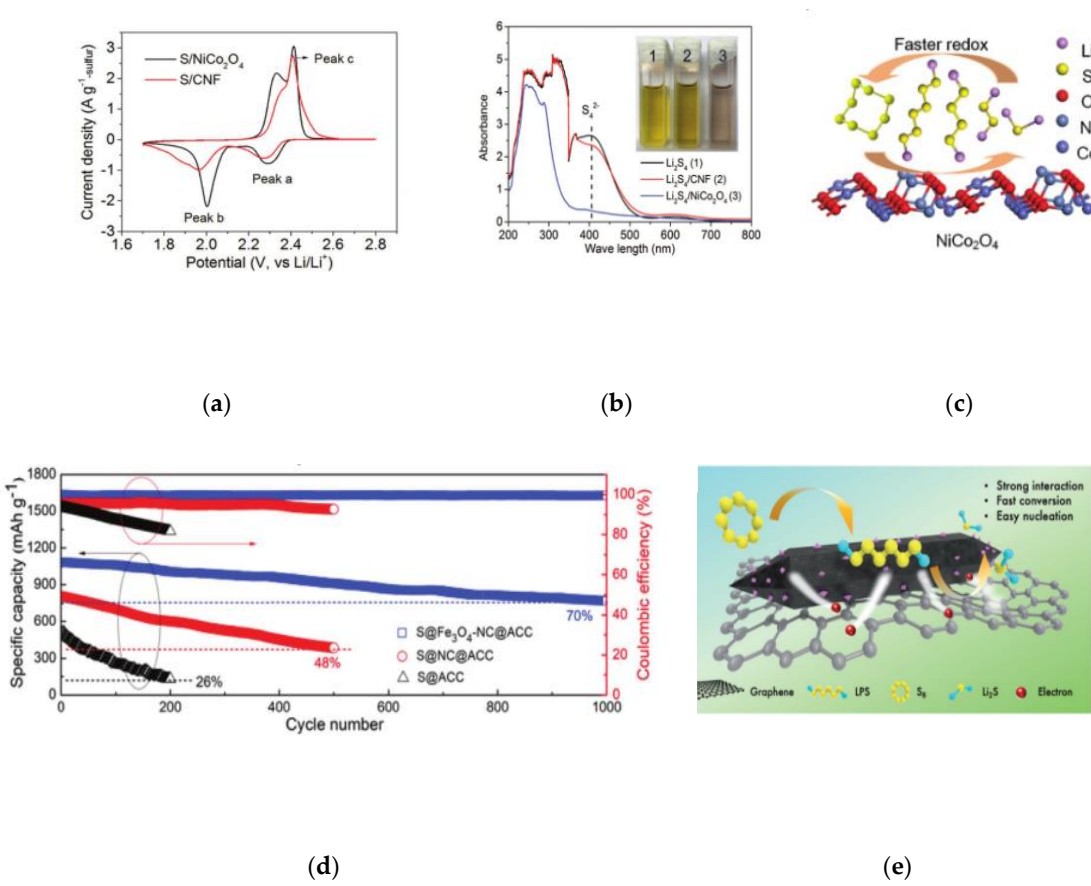

(a)

(b)

(c)

(d)

(e)

**Figure 4.** *Cont.*

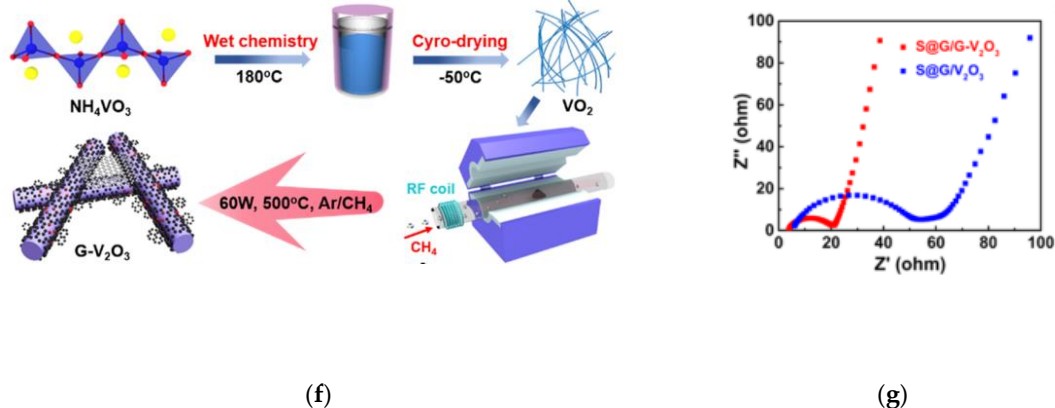

<div align="center">(<b>f</b>)　　　　　　　　　　　　　　　(<b>g</b>)</div>

**Figure 4.** (**a**) CVs of the S/NiCo$_2$O$_4$ and S/CNF composites at the scan rate of 0.1 mV s$^{-1}$ [34]. (**b**) Demonstration of the static adsorption of Li$_2$S$_4$ on NiCo$_2$O$_4$ nanofibers and CNFs. UV–Vis spectra of supernatant of Li$_2$S$_4$ solution after the adsorption test, and the corresponding photos (inset) [34]. (**c**) A schematic showing the faster redox kinetics of LiPS on the NiCo$_2$O$_4$ surface [34]. (**d**) Cyclic stability and Coulombic efficiency (at 0.2 C) for sulfur impregnated in different composite cathodes as noted [35]. (**e**) Effective anchoring of polysulfides and promoting their conversion by Fe$_3$O$_4$/NC/G composites [36]. (**f**) Schematic illustration of the synthetic procedure of G-V$_2$O$_3$ [37]. (**g**) EIS curves of S@G/G-V$_2$O$_3$ and S@G/V$_2$O$_3$ cathodes with a sulfur mass loading of 3.6 and 3.5 mg cm$^{-2}$, respectively [37].

### 3.2. Sulfides

Metal sulfides, such as NiS, CoS, and VS$_2$ [38–40], have attracted special attention due to their strong chemical adsorption and the better conductivity, as well as various thermodynamically stable crystal structures and stoichiometric compositions. For this reason, Chunlong Dai has designed a new type of hollow Co$_9$S$_8$ tube as the host for sulfur (Figure 5a) [41]. With great chemical adsorption and catalytic ability, Co$_9$S$_8$ helps to immobilize polysulfides and catalyze the conversion of LPSs so as to prolong the cycle life of the battery. The hollow structure can further prevent polysulfide from escaping to the anode, and provide enough space to accommodate the necessary volume expansion. Besides, the arranged tubular array provides a pipeline for the rapid conduction of electrons and lithium ions, thus increasing the conductivity. The high discharge capacity and low reduction polarization show that Co$_9$S$_8$ is helpful to promote the electrochemical reaction kinetics in the discharge/charge process (Figure 5b). After 600 cycles at 1 C, the capacity decay rate is only 0.026%.

However, in the above research, the sizes of metal sulfides particles are usually large. In order to prepare the cathode with high sulfur loading and high energy density, the design of nanostructure needs to be optimized. Wanlong Li and co-workers designed a kind of layer-by-layer electrode structure [42]. As shown in Figure 5c, each layer of electrode is composed of ultrafine CoS$_2$ nanoparticles uniformly growing on both sides of reduced graphene oxide (rGO) embedded in porous carbon. Therefore, the interconnection conductive frame with layered pore structure is beneficial to increase its conductivity. At 1.0 and 5.0 C, the electrode has excellent discharge performance of 1180.7 and 700 mAh g$^{-1}$, respectively. At 5.0 C, it still keeps good cycle stability after 1000 cycles and the capacity attenuation of each cycle is 0.033%.

### 3.3. Nitrides

As for transition metal nitrides, the conductivity is higher than those of metal oxides and sulfides. Through strong chemical interaction, the metal ions and electron-rich nitrogen can help to restrict both lithium and polysulfide ions. In addition, the electron transfer between the polar metal nitride and polysulfide accelerates the reversible conversion of the intermediate products during the charging and discharging process, thus reducing the shuttle effect [43]. Therefore, metal nitrides have been widely studied in recent years. For example, Linlin Zhang and co-workers introduced indium nitride

(InN) nanowires into lithium–sulfur batteries [44]. Indium ions and electron-rich nitrogen atoms in InN have strong chemical adsorption on polysulfides, while the fast electron transfer on the surface of InN accelerates the conversion of polysulfide in the working cell. For this reason, the lithium sulfur battery with InN modified separator has excellent rate performance and long stable cycle life. As shown in Figure 5d, the capacity decay rate of each cycle after 1000 cycles is only 0.015%. Furthermore, Kuikui Xiao and co-workers prepared a 2D nitrogen doped carbon structure with uniform distribution of Co$_4$N particles (MOF-Co$_4$N) [45]. The porous structure of N-doped carbon interconnection network is conducive to electronic transport and ion diffusion. Additionally, Co$_4$N nanoparticles provide strong chemical anchoring for polysulfides. The CV curve shows that Co$_4$N nanoparticles can catalyze the conversion of Li$_2$S$_6$ to Li$_2$S$_2$ and Li$_2$S, reducing the polarizability (Figure 5e).

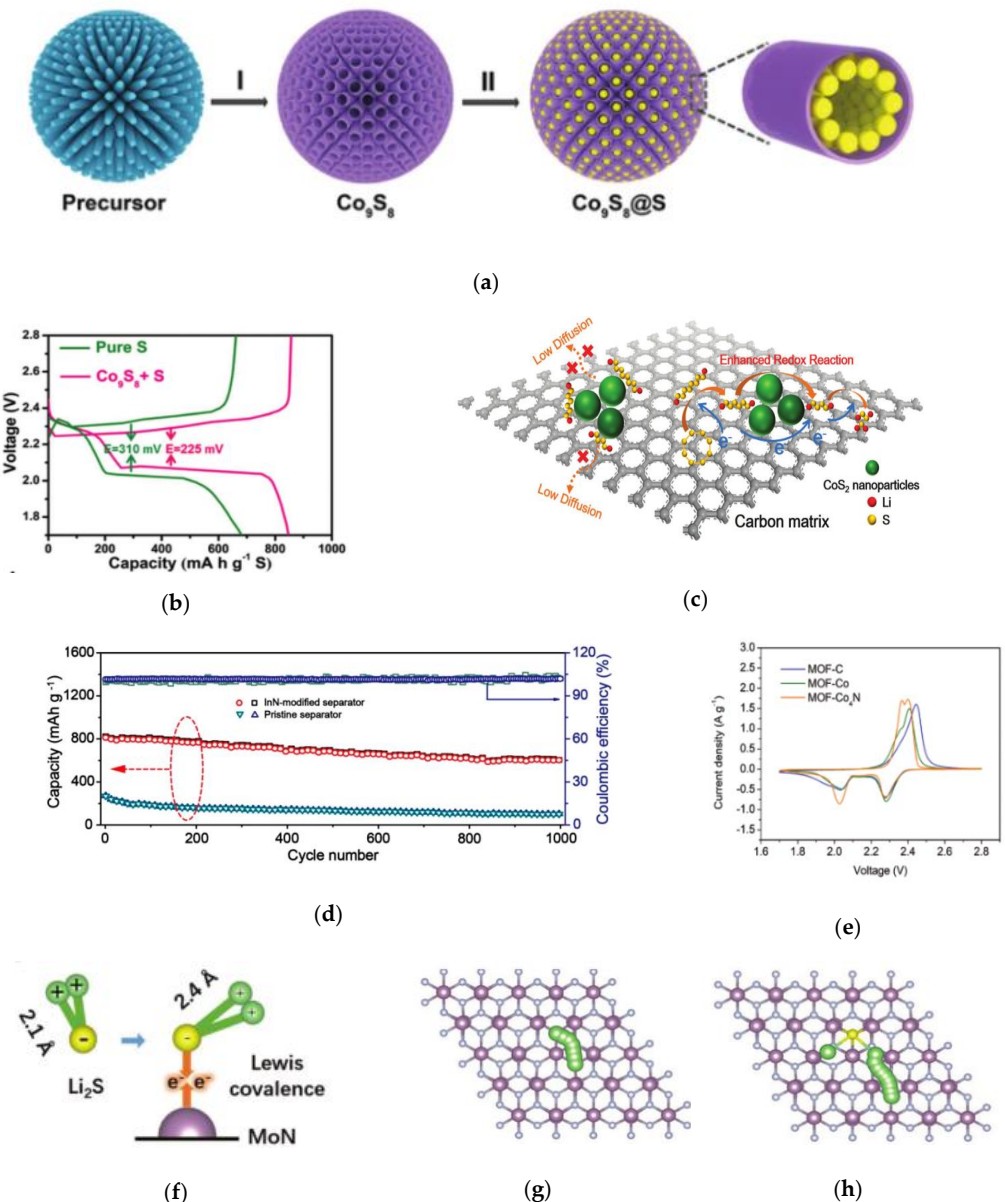

**Figure 5.** (**a**) Schematic illustration of the fabrication of S@Co$_9$S$_8$ composites [41]. (**b**) Discharge–charge curves of S and Co$_9$S$_8$+S electrodes [41]. (**c**) The role of CoS$_2$ nanoparticles on polysulfides capture and conversion [42]. (**d**) Long-term cycling performance of Li−S batteries at 1.0 C after 1000 cycles [44]. (**e**) The second cycle of CV [45]. (**f**) Representation of the covalence-activation mechanism on MoN [46]. (**g**) Li ion diffusion path and (**h**) Li$_2$S decomposition path on MoN. The Li, S, Mo, and N atoms are denoted by green, yellow, purple, and gray balls, respectively [46].

Da Tian and co-workers developed a separator material of MoN-G/PP [46]. Figure 5f shows the adsorption activation of $Li_2S$ on MoN. Figure 5g,h shows that the Li ions generated on the MoN move along the same path as the adsorbed isolated Li, confirming the relationship between the decomposition of $Li_2S$ and the diffusion of Li ions. The decomposition activation energy of $Li_2S$ on MoN is significantly higher than that the diffusion barrier of Li, and the fracture of Li-S bond is the rate determining step. These results show that MoN promotes the decomposition of $Li_2S$ not only through rapid lithium ion transport, but also an effective catalytic pathway. Metal nitrides have high conductivity and can effectively adsorb and catalyze the conversion of polysulfides. However, the preparation of metal nitrides often requires high temperature treatment in an $NH_3$ atmosphere and the active surface area needs to be optimized so as to improve the catalytic efficiency.

### 3.4. Phosphides

Transition metal phosphide is an efficient electrocatalyst with simple and mild synthesis process. Due to its high conductivity compared with oxide and sulfide, it has been widely studied in recent years. Huadong Yuan and co-workers found that the transition metal phosphide can not only capture soluble polysulfide, but also effectively catalyze the decomposition of $Li_2S$ and improve the utilization of active substances [47]. It is found that the capture ability of $Fe_2P$ to polysulfide is better than that of $Ni_2P$ and $Co_2P$, which leads to the accumulation of polysulfide on the cathode surface and hinders the subsequent diffusion and transformation of polysulfide. Therefore, the cycle performances of $Ni_2P@NPC$ and $Co_2P@NPC$ are better than that of $Fe_2P@NPC$.

Shaozhuan Huang and co-workers proposed a new type of nanometer iron phosphide catalyst for lithium sulfur battery [48]. As shown in Figure 6a, the FeP nanocrystals provide efficient chemical adsorption of polysulfides through the enhanced bond formed by Li–P and Fe–S bonds. The FeP nanocrystals fixed on the rGO-CNT framework provide a large number of interfaces for the nucleation and growth of $Li_2S$, thus accelerating the redox reaction kinetics. With the same nanotube morphology, $CF/FeP@C@S$ has better electrochemical performance compared with $CF/Fe_3O_4@C$. Jiadong Shen and co-workers found that according to the calculation results of first principles density functional theory (DFT), the different performance between $Fe_3O_4$ and FeP is mainly due to the shift of P band to Fermi level, which can adjust the Fermi level of interface electrodynamics by increasing the electronic concentration of polysulfides (Figure 6b) [49]. Yi Chen and co-workers prepared a kind of homogeneous Co-Fe-P nanocubes with highly interconnected pore structure [50]. Figure 6c,d shows that Co-Fe-P has strong polysulfide capture and catalytic ability than Co-Fe, which result in the high specific capacity, excellent rate performance, and long cycle. For metal phosphides, their high conductivity and strong chemical interaction with polysulfides are conducive to the formation of $Li_2S$.

The above research showed that the metal compound not only has strong adsorption, but also a certain catalytic effect on LPSs. However, due to the limited conductivity of metal compound, the resistance of the Li-S cell will increase, and the captured LPSs could not be catalyzed in time, which stay on the surface of the polar material, result to the inadequate utilization of the captured active material and the decrease of rate performance. Therefore, research has gradually transitioned from poorly conductive oxides to better conductive phosphides. Besides, metal compounds are often compounded with conductive materials such as carbon materials to improve conductivity and the rate performance of batteries.

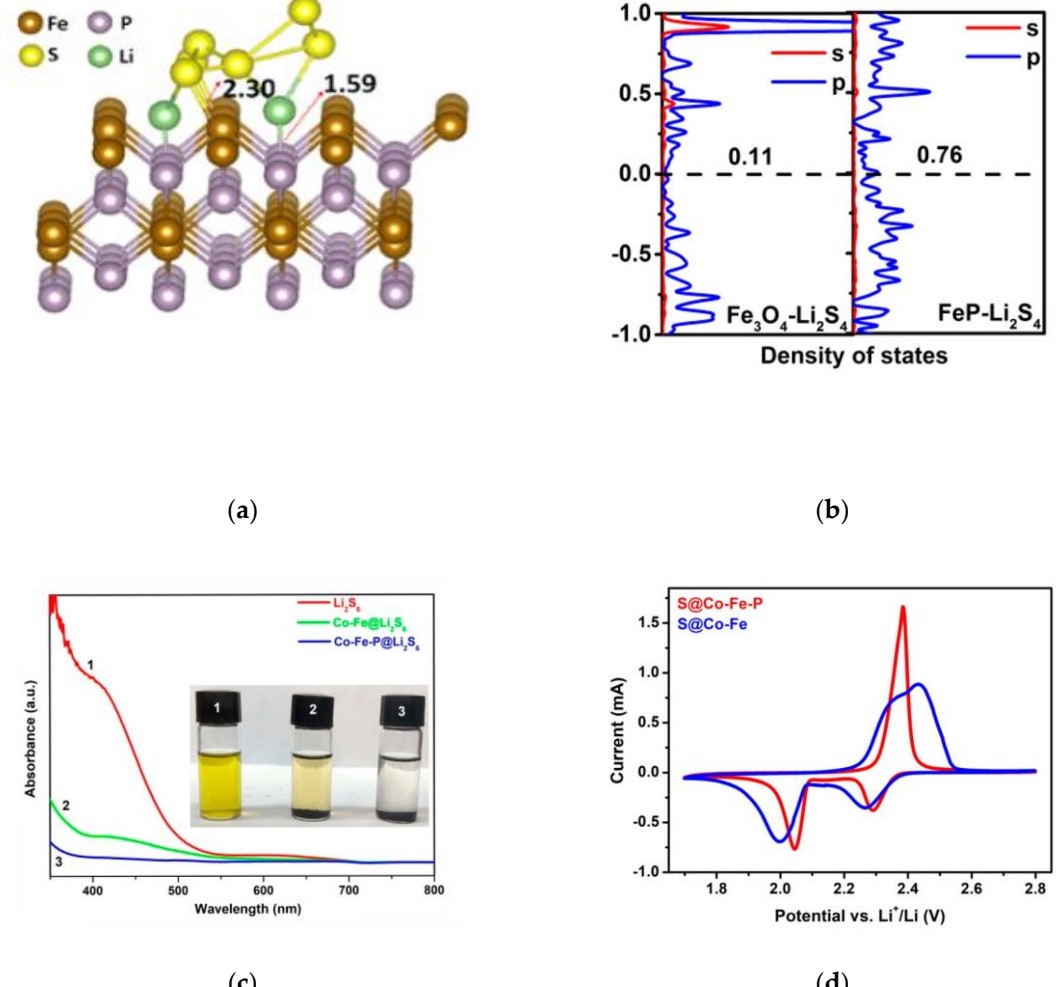

(**a**)

(**b**)

(**c**)

(**d**)

**Figure 6.** (**a**) Geometry of Li$_2$S$_6$ binding to the (111) plane of FeP [48]. (**b**) Density of states analysis of s bands and p bands of Li$_2$S$_4$ in Fe$_3$O$_4$(111)−Li$_2$S$_4$ and FeP(211)−Li$_2$S$_4$ systems [49]. (**c**) UV−Vis adsorption spectra of a Li$_2$S$_6$ solution before and after the addition of Co−Fe or Co−Fe−P powder. The inset shows the digital photo of Li$_2$S$_6$ solution before and after the addition of Co−Fe or Co−Fe−P powder [50]. (**d**) The second-cycle CV curves of the S@Co−Fe−P and S@Co−Fe cathodes at 0.1 mV s$^{-1}$ [50].

## 4. The Second-Generation Catalyst Materials

Non-conductive materials have excellent anchoring ability to polysulfides, but the conversion efficiency is limited. By contrast, although the specific conductive materials have poor anchoring ability, their conversion ability is good. Therefore, it is a feasible way to suppress the shuttle effect and improve the electrochemical performance by combining some materials with high adsorption and catalysis ability to prepare a type of composite with fixation and conversion ability on polysulfides (Figure 1d). This heterostructure is mainly composed of metal oxides, sulfides, or nitrides, such as Co9S8–CoO [51], NiO–NiCo2O4 [52], and MoN-VN [53].

Tianhong Zhou and co-workers reported a TiO$_2$–TiN heterostructure [54], which combines the advantages of high adsorption of TiO$_2$ and high catalysis and conductivity of TiN to realize the capture-diffusion-conversion of polysulfides at the interface (Figure 7a). During 2000 cycles at 1 C, when the sulfur content was 3.1 and 4.3 mg cm$^{-2}$, the capacity retention rate was 73% and 67%, respectively (Figure 7b). Yingze Song and co-workers found that the key to improving the electrochemical performance of lithium sulfur battery is to establish a balance between immobilization and conversion by adjusting the mass ratio of VO$_2$ and VN [55]. Rongrong Li and co-workers proposed a kind of MoO$_2$ and Mo$_3$N$_2$ heterostructure material with porous nano band morphology and low

specific surface area (95 m$^2$ g$^{-1}$), which makes the conformal deposition of S/Li$_2$S have a uniform spatial distribution [56]. TG curves show that it can achieve high S loading of 75 wt% even at low specific surface area (Figure 7c). Under high sulfur load of 3.2 mg cm$^{-2}$, after 1000 cycles, the specific capacity is 451 mAh g$^{-1}$ (Figure 7d). Besides, the affinity between polysulfides and the heterostructure makes it possible to obtain dendritic free Li coating on the lithium anode even after long-term cycling. Jinlin Yang and co-workers prepared a new MoO$_2$-Mo$_2$N binary nano-belt as separator coating material by in situ topological nitriding method (Figure 7e) [57], which not only acts as a physical barrier to reduce the shuttle effect, but also combines the polarity of MoO$_2$ and the conductivity of Mo$_2$N, regulating the conversion of polysulfide and optimizing the nucleation of solid Li$_2$S during the cycling process (Figure 7f).

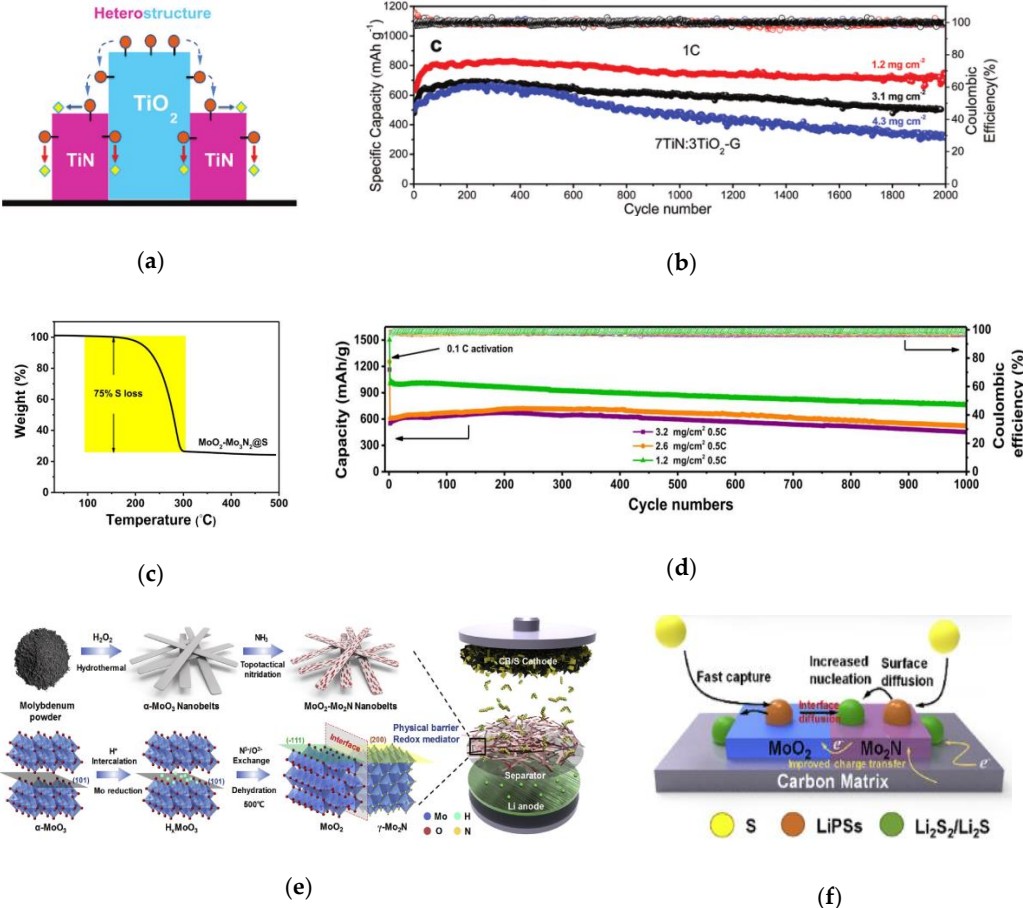

**Figure 7.** (**a**) The TiO$_2$–TiN heterostructure surface [54]. (**b**) Ultra-long cycling for the optimal cells with the 7TiN:3TiO$_2$–G coating layer at 1 C for about 2000 cycles with different sulfur loadings of 1.2, 3.1, and 4.3 mg cm$^{-2}$ [54]. (**c**) TG measurement curve of MoO$_2$–Mo$_3$N$_2$@S composite [56]. (**d**) Discharge capacity and Coulombic efficiency as a function of cycling number for MoO$_2$–Mo$_3$N$_2$@S cathodes based on different sulfur loadings at 0.5 C for 1000 cycles [56]. (**e**) Schematic illustration and the atom model of the synthesis process of MoO$_2$–Mo$_2$N heterostructure [57]. (**f**) Schematics of LiPSs species conversion and Li$_2$S nucleation process on the surface of bare MoO$_2$–Mo$_2$N binary structure [57].

The heterojunction combines high absorption, catalytic, and conductive materials to accelerate the absorption and diffusion of lithium polysulfide, thereby enabling more efficient catalysis and improving reaction kinetics. In the above research, the choice of the second generation catalyst materials is often connected with semiconductors, so its conductivity and catalytic performance are better than those of the first generation catalysts, and the cycle performance and rate performance of lithium–sulfur batteries are greatly improved.

## 5. The Third Generation Catalyst Materials

With the development of the first- and second-generation catalyst materials, researchers have gradually realized that the effective captureability for polysulfides and the conductivity of the material are the key factors for catalyst materials to inhibit the shuttle effect. For example, in the first generation of materials, catalysts have improved from metal oxide to metal phosphide with better conductivity; in the second generation, the heterostructures formed by the combination of non-conductive and conductive materials have been prepared. Therefore, in the subsequent design of catalytic materials, many methods have been tried to consciously improve the conductivity of materials. In the research process, the specific surface area of the catalytic materials is intentionally increased to achieve the rapid capture for polysulfides, which can effectively improve the catalytic activity and efficiency, thus reducing the amount of catalyst needed to improve the energy density of the electrode. As a result, a third generation of catalysts with stronger conductivity, faster polysulfide adsorption, and higher catalytic efficiency has emerged, such as metal cobalt, metal carbide, MXene, and single-atom catalysts (Figure 1e).

### 5.1. Metal Cobalt

It was found that the porous Co-N-C is rich in nitrogen doping. When Co-N-C was applied to lithium–sulfur battery, the doped nitrogen site can combine with polysulfide [58,59], the adjacent Co site can be used as the chalcogenide and promote the redox kinetics of polysulfide conversion [60,61], and the porous carbon substrate can also physically capture some polysulfide, thus achieving rapid charge/material transfer.

Wen Hu designed a new type of macroporous/mesoporous Co-N-C structure as separator coating (Figure 8a) [62]. Mesoporous structure and high specific surface area provide abundant exposed adsorption sites (Lewis base N-doping and Lewis acid Co cation) on the interface, which can selectively capture LPSs, thus accelerating the formation of $Li_2S$. The initial capacity of MWCNT/S-70 wt% increases significantly at both low and high current density (1406 mAh $g^{-1}$ at 0.2 C and 1203 mAh $g^{-1}$ at 1 C). The material has nearly 100% coulomb efficiency and high reversible capacity. In order to further improve the conductivity of the catalytic materials, we compounded nitrogen-rich MOFs nanosheets, bio-MOF-100, with carbon nanotubes, doped with cobalt and then carbonized in an inert atmosphere to form dispersed cobalt nanoparticles and in situ auto catalytic carbon nanotubes. In this way, we successfully prepared Co/NCNS/CNT nanocomposites as separator of batteries (Figure 8b) [63]. The three-dimensional conductive network provides a fast path for electron transmission, reducing the resistance of charge transfer (Figure 8c), and the doping of cobalt and nitrogen as well as the nanostructure contributes to accelerating the adsorption on LPSs. In addition, cobalt nanoparticles and Co-N functional groups with uniform distribution are conducive to catalyze the conversion of LPSs to short-chain $Li_2S$. The results show that the shuttle effect of polysulfides is effectively inhibited with the improvement of reaction kinetics. At high sulfur load of 5 mg $cm^{-2}$, the specific capacity can still be kept at 522.1 mAh $g^{-1}$ after 500 cycles under 1 C.

The above results indicated that N-doped graphene can effectively prevent the aggregation of Co particles, and Co-N-C greatly improves the electrochemical redox reaction kinetics. More importantly, these metal Co have high conductivity, which is helpful to realize the rapid charge transfer, leading to significant improvement in the utilization of sulfur, higher capacity, and cycle stability of the batteries. In addition, by increasing the active surface area of the catalysts, the utilization rate of the metal catalysts is improved.

### 5.2. Metal Carbide

Transition metal carbides, such as $W_2C$, $Ti_3C_2$ (MXene), and $Fe_3C$, have attracted extensive attention in the field of lithium sulfur batteries due to their great thermal stability, conductivity, and metal characteristics, and strong adsorption and catalysis conversion ability for polysulfides [64–66].

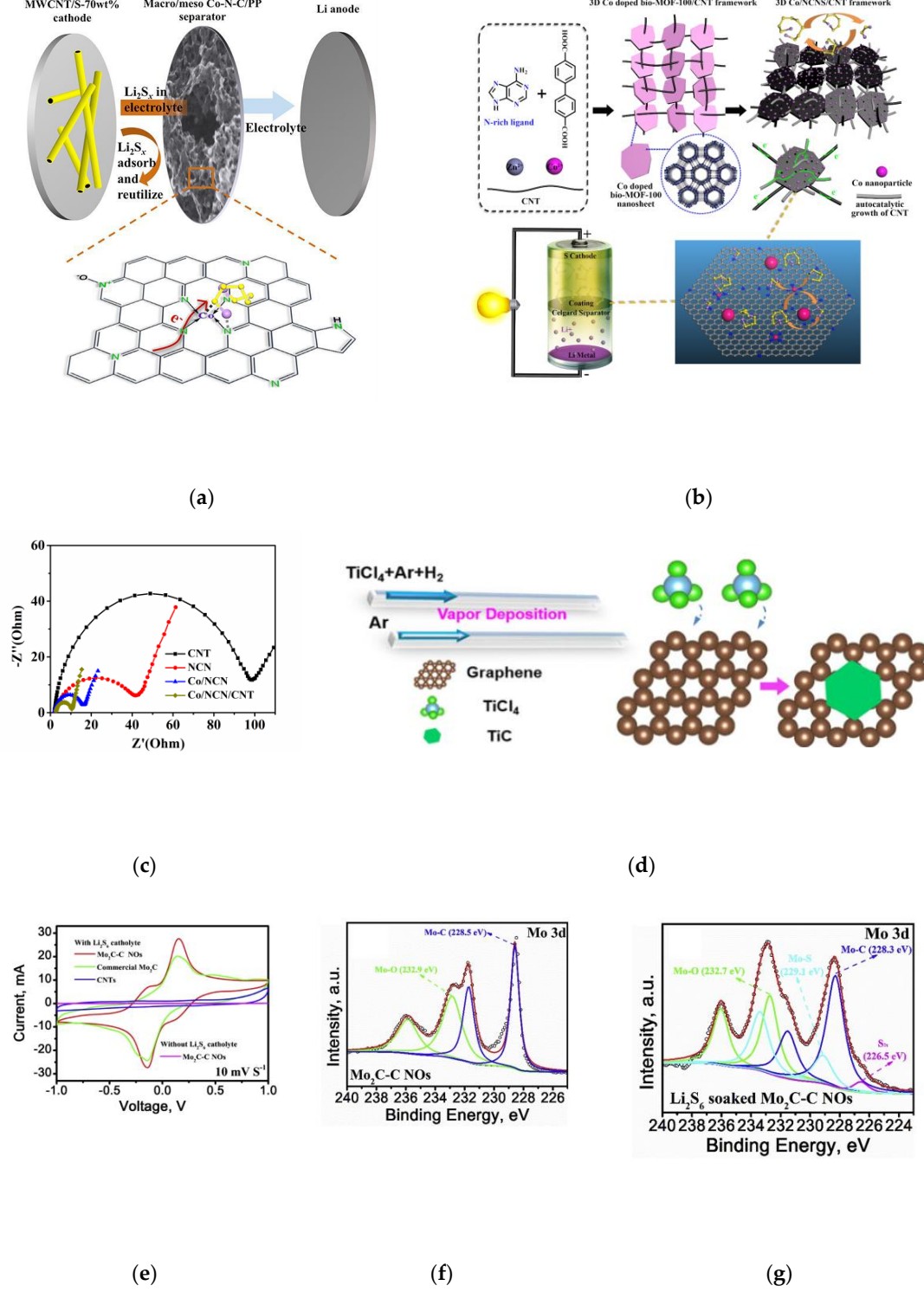

**Figure 8.** (**a**) Schematic configuration of the assembled Li-S cell model and the electrochemical improvement mechanism of the macro/meso Co-N-C-decorated separator [62]. (**b**) The scheme of rational design of 3D Co/NCNS/CNT as separator coatings for high-performance Li-S battery [63]. (**c**) The EIS spectra of different materials in a symmetric cell with $Li_2S_6$ in the electrolyte [63]. (**d**) Schematic illustration for the synthesizing TiC nanosheets directly inlaid into the plane of graphene templates using $TiCl_4$ as the reactant [67]. (**e**) The polarization curves of the symmetrical cells with and without $Li_2S_6$ recorded from −1.0 to 1 V at 10 mVs$^{-1}$ [68]. (**f**) Mo 3d spectra of the $Mo_2C$–C NOs [68]. (**g**) Mo 3d spectra of $Li_2S_6$ soaked $Mo_2C$–CNOs [68].

With graphene as template, Tianhong Zhou and co-workers prepared the planar heterostructure of graphene and titanium carbide (TiC) by direct reaction of carbon source with $TiCl_4$ (Figure 8d) [67]. This heterostructure is beneficial to reduce the diffusion potential barrier of lithium ion and electron. Its cycle performance and magnification performance prove that the planar G-TiC heterostructure has high capture ability and conversion efficiency for polysulfides. Guilin Chen and co-workers prepared $Mo_2C$-C nano octahedron as the sulfur host [68]. As a result, high polarity $Mo_2C$ could adsorb polysulfide by strong Mo-S bond (Figure 8f,g). In addition, $Mo_2C$ nanoparticles with good electrocatalytic activity can promote the redox reaction of polysulfide (Figure 8e). In this method, at 0.1 C, the $Mo_2C$-C NOs@S with 72.15 wt% of sulfur has the specific capacity of 1396 mAh $g^{-1}$. The good conductivity of transition metal carbides not only accelerates the diffusion of electrons and lithium ions, but also promotes the fast conversion of polysulfides, which result in the higher cycling performance of Li-S battery.

### 5.3. MXene

Two-dimensional transition metal carbides and nitrides (the general formula is $M_nC_{n-1}/M_nN_{n-1}$ (n = 2–4)) are called MXene, where "M" refers to transition metals, such as Ti, V, Nb, and Mo [69–71]. MXene are similar to graphene in structure, but they have many functional groups on the surface, such as OH and F, which makes them different from graphene. Because of the stable multilayer structure, large number of surface functional sites and high electronic conductivity, MXene has a great application prospect in lithium–sulfur batteries.

Xiao Liang and co-workers demonstrated that the metal MXene phases ($Ti_3C_2$ and $Ti_3CN$) could interact with LPSs through the formation of thiosulfate [72]. The thiosulfate/polythioic acid groups formed on MXene nanowafers show that the hydroxyl part of functionalized MXene reacts with polysulfide by redox reaction. On this basis, Dashuai Wang and co-workers systematically studied the introduction of different functional groups in $Ti_3C_2$ MXenes as the sulfur host for lithium sulfur battery [73]. By comparing the adsorption capacity of $Ti_3C_2T_2$ to polysulfide and the electrocatalytic performance of $Li_2S$ decomposition, it is found that S and O are the best choice for surface modification of $Ti_3C_2$.

An insurmountable obstacle for MXene nanowafers is the large numbers of hydrogen bonds produced by the surface functional groups. MXene nanowafers have a strong tendency of reaggregation, which results in a large amount of loss of active regions. As a result, the long-term cycling and magnification performance of MXene/S anode are poor. Therefore, before sulfur loading, mesoporous carbon, r-GO, and CNT are coated or two-dimensional nanosheets are integrated into three-dimensional structure to prevent MXene nanosheets from aggregating [74]. Xiaotian Gao first reported the growth of $TiO_2$ quantum dots (QDs) on ultra-thin MXene ($Ti_3C_2Tx$) nanosheet by cetyltrimethyl ammonium bromide-assisted solvothermal synthesis (Figure 9a) [75]. $TiO_2$ QDs are uniformly modified on MXene nanosheets in order to act as a spacer to prevent them from reassembling, thus maintaining the stability of its two-dimensional geometry. In this way, the adsorption capacity of polysulfides enhances and the dynamics of redox accelerates. At 2 C, the high capacity of 680 mAh $g^{-1}$ was maintained after 500 cycles and the attenuation rate per cycle was only 0.04% (Figure 9b).

With multi-layer structure, a large number of surface functional sites and high electronic conductivity, MXene can effectively adsorb polysulfides and catalyze their conversion. Considering the easy aggregation of MXene nanosheet, some spacers or coatings are inserted into MXene, make the two-dimensional nanosheets into a three-dimensional structure to prevent the aggregation, thus improving the cycle performance and rate performance of lithium–sulfur batteries.

### 5.4. Single-Atom Catalyst

As the catalytic performance is related to the particle size, it is assumed that the maximum catalytic efficiency can be achieved with single atom. In fact, single-atom catalysts (SACs) have the advantages of both heterogeneous and homogeneous catalysts, and their theoretical atom utilization rate reaches

up to 100% because of high conductivity. Therefore, it has been used to improve the electrochemical performance of lithium–sulfur battery, and the results show that its catalytic performance is far better than that of traditional metal nanoparticles. However, due to the high surface free energy of the single metal center, single-atom catalysts are chemically unstable and even tend to aggregate into metal nanoparticle, which reduces the catalytic activity and slows down the conversion of lithium polysulfide in electrochemical process. Therefore, in order to stabilize the single-atom catalyst and further improve its catalytic performance, nitrogen-doped graphene with large specific surface area and high conductivity is selected as an ideal substrate material, because the nitrogen group on graphene can interact with transition metal through M-N structure to improve the catalytic activity of monoatomic catalyst in electrochemical process [76].

Zhenzhen Du prepared nitrogen-doped graphene (Co-N/G) by impregnation method to prepare the SACs as the host of sulfur [77]. It was found that the Co-N-C coordination center, as a dual-function electrocatalyst, can promote the formation and decomposition of $Li_2S$ in the process of discharging and charging, respectively. As a result, when sulfur loading is 6.0 mg cm$^{-2}$, the area capacity is 5.1 mAh cm$^{-2}$ after 100 cycles at 0.2 C and the attenuation rate per cycle is 0.029%. Because the preparation of SACs by impregnation depends on the absorption sites of the substrate, the types and yields of SACs are limited. In order to solve the above problems, Guangmin Zhou and co-workers synthesized single-atom graphene with controllable quantity, controllable load, and adjustable composition by crystal seed method (Figure 9c) [78]. Figure 9d shows that the multifunctional SACs have the advantages of strong chemical adsorption on $Li_2S_6$, promotion of conversion between sulfur/lithium polysulfide/$Li_2S$ and the ability to control deposition position of $Li_2S$ (Figure 9e), which contributes to achieving high capacity, rapid conversion, and stability. However, the use of SACs in cathodes usually requires high loading capacity of SACs in active materials, which reduces the utilization of SACs. As the separator coating can reduce the catalyst load, it is a better choice to apply SACs as separator coating in lithium sulfur battery. Linlin Zhang and co-workers prepared a type of nitrogen-doped graphene (Ni@NG) with Ni-N$_4$ structure by pyrolysis [79]. Acts as a polysulfide trap, the oxidized Ni site in the Ni-N$_4$ structure can catalyze reversibly the conversion of polysulfides by forming a strong $Sx^{2-}$ Ni-N bond (Figure 9f). In addition, on Ni@NG, the charge transfer between LPSs and nickel makes it have lower free energy and decomposition energy barrier in the electrochemical process, which accelerates the dynamic transformation of LPSs in the charge and discharge process. In this method, the lithium–sulfur battery has excellent rate performance and stable cycle life with capacity attenuation rate of 0.06% per cycle. Kun Zhang and co-workers found that Fe-SACs have better performance in lithium sulfur battery compared with three other catalysts Fe, Co, and Ni-SACs [80]. Even under the low metal loading of about 2 µg, after 750 cycles at 0.5 C, the reversible specific capacity of the cell with high sulfur loading is 891.6 mAh g$^{-1}$, and the capacity retention rate is 83.7%. The single atoms with atomic size and single metal center have the largest atom utilization rate, unsaturated metal sites and unique electronic structure, which can effectively adsorb and catalyze the redox kinetic of LPSs and improve sulfur utilization rate and catalytic efficiency of the Li-S battery.

In a word, the third-generation catalyst has the characteristics of strong adsorption and high catalytic performance on LPSs, while the electrical conductivity is superior to that of the first- and second-generation catalyst materials. In terms of catalyst utilization, the third-generation catalyst has significantly improved because it has a high specific surface area of electrochemical activity, and research also gradually transition from metal cobalt with low catalyst utilization to single atom catalyst with high utilization, resulting in high sulfur utilization. Using the third-generation catalyst, people manage to improve the rate performance, cycle performance, and energy density of the battery.

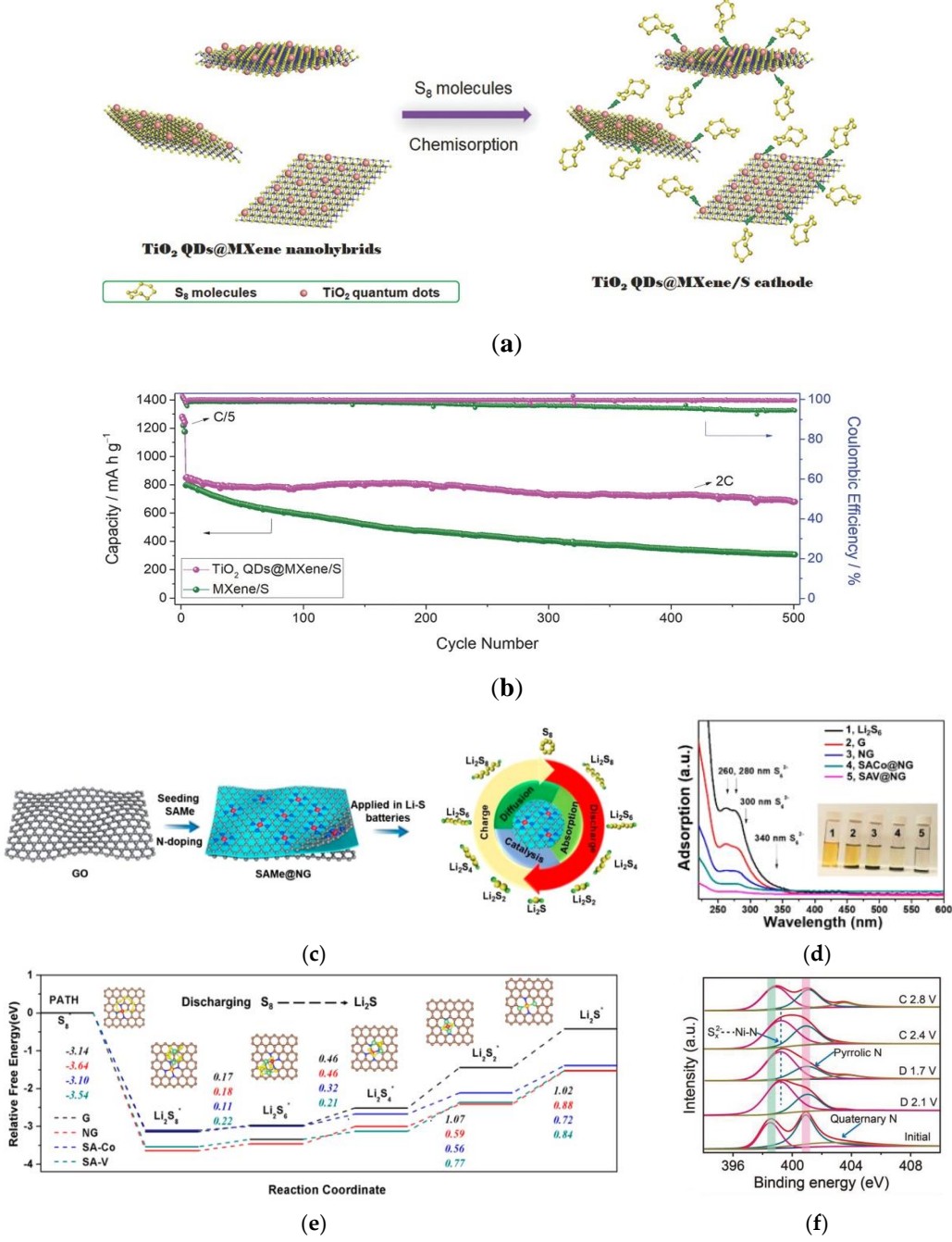

**Figure 9.** (**a**) Schematic configuration of the assembled Li-S cell model and the electrochemical improvement mechanism of the macro/meso Co-N-C-decorated separator [75]. (**b**) The scheme of rational design of 3D Co/NCNS/CNT as separator coatings for high-performance Li-S battery [75]. (**c**) The EIS spectra of different materials in a symmetric cell with $Li_2S_6$ in the electrolyte [78]. (**d**) Schematic illustration for the synthesizing TiC nanosheets directly inlaid into the plane of graphene templates using $TiCl_4$ as the reactant [78]. (**e**) The polarization curves of the symmetrical cells with and without $Li_2S_6$ recorded from $-1.0$ to 1 V at 10 mVs$^{-1}$ [78]. (**f**) Mo 3d spectra of the $Mo_2C$–C NOs.

## 6. The Future Catalyst Materials

It is found that the key to inhibiting the shuttle effect is to improve the ability of capturing polysulfides and the rate of transition between LPSs and short-chain $Li_2S_{(2)}$. The catalyst with high catalytic effect on LPSs in lithium–sulfur battery can effectively inhibit the shuttle effect. Based on

the structure and properties of the reported catalysts, we can find that the design of highly efficient catalytic materials needs to consider not only strong chemical adsorption on polysulfides, but also the conductivity, catalysis, and mass transfer. As shown in Figure 10, the design of a catalyst with high catalytic performance requires a combination of the following; (1) the catalytic sites are uniformly dispersed and have high catalytic activity; (2) the catalysts are lipophilic and sulfuric nanocomposites with fast lithium ion diffusion and fast absorption for polysulfides; and (3) the catalysts should have high conductivity, which provides an electron transfer channel for the rapid occurrence of electrocatalysis. In addition, due to high electrochemically active surface area of catalysts, 2D materials and 3D materials with network structure are needed to improve catalytic activity and efficiency. As a result, this not only reduces the amount of catalyst needed to improve the energy density of the electrode, but also provides a larger specific surface area for the uniform adsorption of polysulfides and the deposition of discharge products of $Li_2S_2/Li_2S$.

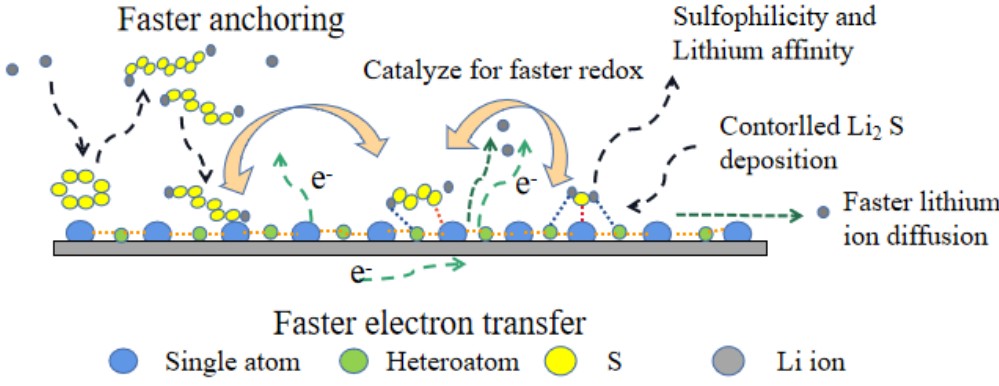

**Figure 10.** The next-generation catalyst material for LPSs.

## 7. Conclusions

In this review, the development of catalytic materials for LPSs in Li-S battery is divided into three generations. The conversion reaction involving multiple electrons requires rapid electron and lithium-ion transport, while retarding polysulfide dissolution. Additionally, utilizing reaction pathways with low activation barrier for the conversion of LPSs contributes to preventing the shuttle effect. It can be concluded that the development of catalytic materials for lithium sulfur battery is related to the ability of polysulfide capture, conductivity, catalysis, and mass transfer. Therefore, the design of catalytic materials for high-performance lithium sulfur battery needs high conductivity, fast polysulfide capture ability, strong sulfur and lithium affinity, small catalytic particles, and large specific surface area of electrochemical activity. Catalytic materials with high conductivity, both lipophilic and thiophile sites will become the next generation catalytic materials, such as heterosingle atom catalysis, heterometal carbide. The development of these catalytic materials will help catalyze LPSs more efficiently and improve the reaction kinetics, thus providing guarantee for lithium sulfur batteries with high load or rapid charge and discharge, which will promote the practical application of lithium–sulfur battery.

**Author Contributions:** Investigation, H.-J.Z., C.-L.S.; writing—original draft preparation, H.-J.Z., C.-L.S.; writing—review and editing, L.-P.S., X.-J.H. and Y.-P.C.; funding acquisition, X.-J.H. and Y.-P.C., H.-J.Z. and C.-L.S. contributed equally. All authors have read and agree to the published version of the manuscript.

**Funding:** This research was funded by [National Natural Science Foundation of China] grant number [21471061, 21671071]; [Science and Technology Planning Project of Guangdong Province, Guangzhou, China] grant number [2015B010135009, 2015B010135009, 2019A050510038]; [innovation team project of Guangdong Ordinary University] grant number [2015KCXTD005]; [the great scientific research project of Guangdong Ordinary University] grant number [2016KZDXM023]; [Guangdong Natural Science Foundation Project] grant number [2019A1515010841].

**Acknowledgments:** This work was supported by the National Natural Science Foundation of China (Grant No.21471061 and 21671071); the Science and Technology Planning Project of Guangdong Province, Guangzhou, China (No. 2015B010135009, 2017B090917002, 2019B1515120027 and 2019A050510038); innovation team project of Guangdong Ordinary University (No. 2015KCXTD005); and the great scientific research project of Guangdong Ordinary University (No. 2016KZDXM023). Guangdong Natural Science Foundation Project (No. 2019A1515010841).

**Conflicts of Interest:** The authors declare no conflict of interest.

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
