# Peer review of "The Development of Catalyst Materials for the Advanced Lithium–Sulfur Battery"

_catalysts, doi:10.3390/catal10060682_

Round 1

Reviewer 1 Report

This work reviews the progress in design of catalyst materials for lithium-sulfur battery as a promising energy storage system. The review is well structured and is well written. It is a nice contribution that meets the standards of catalysts and should be published after minor revisions as below:

It is suggested to include the perspectives and the future resaerch directions of this fields.

Most of the figures need improvement:

  • Figure 2(c) is missing
  • Figure 6c, 6b, 7c, 8c,8e, 9c

Author Response

Point 1: It is suggested to include the perspectives and the future resaerch directions of this fields.

 Response 1: As suggested by the reviewer, the perspectives and the future resaerch directions of this fields was added in the revised manuscript.

Catalytic materials with high conductivity, both lipophilic and thiophile sites will become the next generation catalytic materials, such as heterosingle atom catalysis, heterometal carbide. The development of these catalytic materials will help catalyze LPSs more efficiently and improve the reaction kinetics, thus providing guarantee for lithium sulfur batteries with high load or rapid charge and discharge, which will promote the practical application of lithium-sulfur battery.

Point 2: Most of the figures need improvement: Figure 2(c) is missing; Figure 6c, 6b, 7c, 8c,8e, 9c.

Response 2: As suggested by the reviewer, we have improved the figures as mentioned in the revised manuscript.

Reviewer 2 Report

The authors review the current catalytic materials used in the lithium sulfur battery system. The manuscript can be followed very well and provides an overview of this system and its problems. I think that the authors have done a good job and suggest it to be published in catalyst in the present form.

Author Response

Thanks very much for your kind comments.